

# TiO₂ nanoparticles affect the bacterial community structure and *Eisenia fetida* (Savigny, 1826) in an arable soil

Katia Berenice Sánchez-López[1], Francisco J. De los Santos-Ramos[2], Elizabeth Selene Gómez-Acata[3], Marco Luna-Guido[3], Yendi E. Navarro-Noya[4], Fabián Fernández-Luqueño[1] and Luc Dendooven[3]

[1] Nanoscience and Nanotechnology Program, Centro de Investigación y de Estudios Avanzados del Instituto Politécnico Nacional, Mexico City, Mexico
[2] Department of Physiology, Biophysics and Neuroscience, Centro de Investigación y de Estudios Avanzados del Instituto Politécnico Nacional, Mexico City, Mexico
[3] Department of Biotechnology, Centro de investigación y de estudios avanzados del Instituto Politécnico Nacional, Mexico City, Mexico
[4] CONACYT Cathedra, Tlaxcala Center of the Behavior Biology, Autonomous University of Tlaxcala, Tlaxcala, Mexico

Corresponding author
Luc Dendooven,
dendooven@me.com

## ABSTRACT

The amount of nanoparticles (NP), such as TiO₂, has increased substantially in the environment. It is still largely unknown, however, how NP might interact with earthworms and organic material and how this might affect the bacterial community structure and their functionality. Therefore, an arable soil was amended with TiO₂ NP at 0, 150 or 300 mg kg⁻¹ and subjected to different treatments. Treatments were soil amended with ten earthworms (*Eisenia fetida* (*Savigny, 1826*)) with fully developed clitellum and an average fresh mass of 0.5 to 500 g dry soil, 1.75 g tyndallized Quaker® oat seeds *Avena sativa* (L.) kg⁻¹, or earthworms plus oat seeds, or left unamended. The bacterial community structure was monitored throughout the incubation period. The bacterial community in the unamended soil changed over time and application of oats, earthworm and a combination of both even further, with the largest change found in the latter. Application of NP to the unamended soil and the earthworm-amended soil altered the bacterial community, but combining it by adding oats negated that effect. It was found that the application of organic material, that is, oats, reduced the effect of the NP applied to soil. However, as the organic material applied was mineralized by the soil microorganisms, the effect of NP increased again over time.

## INTRODUCTION

Nanotechnology has attracted a lot of attention as a source of new materials or devices with specific characteristics that have never been developed before. These materials or devices are small so their possible hazardous effect on the environment and ecosystems in

general is largely unknown (*McKee & Filser, 2016*). This has led to the emergence of nanotoxicology, that is, a field of research that focuses on the effect of nanotechnology on environmental and human health.

Manufactured titanium dioxide nanoparticles ($TiO_2$-NP) have several applications and are used widely (*Hu et al., 2010*). For instance, the production in the USA will reach $2.5 \times 10^6$ tons $y^{-1}$ by 2025 (*McShane et al., 2012*). This nanomaterial and its by-products will ultimately enter the environment (*Joo & Zhao, 2017*; *Zhang et al., 2017b*). However, little information is available on how $TiO_2$-NP might affect ecosystems, such as water, sediments and soil (*Ge et al., 2013*; *Gottschalk et al., 2009*; *Hu et al., 2010*; *Nogueira et al., 2012*; *Van Gestel, Kool & Ortiz, 2010*). Most of the $TiO_2$-NP will end up in soil and there are concerns they might enter the food chain and accumulate in animals and humans (*McKee & Filser, 2016*; *Yeo & Nam, 2013*).

Soil is formed as an interaction of plants, soil microorganisms and the micro- and macrofauna within a physicochemical environment. Contamination with xenobiotic and/or recalcitrant chemicals, such as $TiO_2$-NP, might alter the soil microbial community, and the micro- and macrofauna, such as earthworms. Earthworms (*Oligochaeta*) live in a wide variety of soils and their abundance can reach 300 $m^{-2}$ (*Bardgett & Van Der Putten, 2014*). Earthworms, such as *Eisenia fetida* (Savigny, 1826), play a key role in soil. As they burrow through a soil, earthworms contribute to pedogenesis, water regulation, nutrient cycling, aeration, removal of contaminants and soil structure formation (*Blouin et al., 2013*). Although earthworms accelerate the removal of organic contaminants from soil, these pollutants might inhibit their activity. *E. fetida* can act as a toxicity bioindicator and has been used as a standard animal for toxicology experiments (*OECD, 1984*; *Saint-Denis et al., 2011*). It is sensitive to various toxicants and can be cultured easily in the laboratory. Earthworms are ubiquitous, so they have been used to determine the possible toxicity of soil contaminants, such as $TiO_2$-NP toxicity (*OECD, 1984*).

Soil microorganisms play a crucial role in soil nutrient cycling but only a limited number of studies have been published that investigated the effect of $TiO_2$-NP on C and N dynamics and soil biodiversity (*Hu et al., 2010*; *Nogueira et al., 2012*; *Van Gestel, Kool & Ortiz, 2010*). Most of these studies found an effect of NP on soil biodiversity but the effect depends largely on the type of NP and soil characteristics (*Frenk et al., 2013*). Metal NP are difficult to remove from the environment, so their effect on microbial community structure might be lasting. More experiments are therefore needed to investigate how NP affect the bacterial community structure under different soil conditions so as to increase our understanding of the possible effects they might have on soil fertility (*Schlich, Terytze & Hund-Rinke, 2012*). Most studies investigate the effect of one factor separately on the microbial community structure, but in this study, we investigated how a combination of two factors, that is, earthworms and NP, affected the bacterial community structure over time. Therefore, an arable soil amended with oat seeds *Avena sativa* (L.), *E. fetida* or a combination of both was spiked with different concentrations of $TiO_2$-NP (0, 150 or 300 mg $TiO_2$-NP $kg^{-1}$ dry soil). The number of adults, juveniles, cocoons and shells of *E. fetida* were counted and the soil bacterial community structure monitored during an aerobic incubation of 90 days. The objective of

**Table 1 Some characteristics of the TiO₂-nanoparticles used in this study.**

| Characteristic | |
|---|---|
| Chemical formula | $TiO_2$ |
| Colour | White |
| Density (g cm$^{-3}$) | 4.23 |
| Molecular weight (Da) | 79.87 |
| Melting point (°C) | 1,843 |
| Crystalline phase | Anatase |
| Particle size (nm) | 50–100 |
| Crystallographic system | Hexagonal |
| Magnetic properties | Weakly ferromagnetic |

this study was to determine the effect of $TiO_2$-NP on the bacterial community structure in an arable soil amended with *E. fetida* and supplemented with oats as food.

# MATERIALS AND METHODS

## Nanoparticles characteristics

Titanium dioxide nanoparticles were purchased from "*Investigación y Desarrollo de Nanomateriales S.A. de C.V., México*" (Fig. S1). Some selected characteristics of the $TiO_2$-NP used in this experiment are given in Table 1. No coating or stabilizing agent was used for the $TiO_2$ particles. The composition of the $TiO_2$-NP was determined by analysis of X-ray diffraction with a Philips X'Pert diffraction equipment (Eindhoven, The Netherlands). The morphology and the chemical composition were determined by transmission electron microscopy equipped with the energy dispersive spectroscopic technique, in a Tecnai F30 HRTEM microscope (Tecnai, Hillsboro, OR, USA).

## Earthworms

*Eisenia fetida* was obtained from the "*Universidad Autónoma de Chapingo*" (Mexico State, Mexico). They were kept in pre-composted bedding with kitchen organic wastes at "Cinvestav" (Mexico city, Mexico). The organic waste, that is, mostly oats and melon, was obtained from a household that washed the vegetables and fruits before domestic use so that the discarded organic waste contained a minimum of pesticide or other organic components that might have altered the soil bacterial community. No pesticides or other pest repellents were applied to the organic waste. After 30 days, the earthworms that weighed at least 0.5 g were selected and used in this study.

## Soil characteristics

Soil was collected from an arable field in Acolman (State of Mexico, Mexico, 19°38′56.4″N, 98°54′13.1″W) at 2,260 masl with a mean annual temperature of 15.2 °C. The climate is Cwb (temperate with dry winters) based on the Köppen climate classification system (*Kottek et al., 2006*). The average annual rainfall is 595 mm, mainly from June to September. The application of fertilizer and pesticides was extremely limited as no intensive agriculture takes place in this area.

**Table 2 Treatments applied to the arable soil combining the application of oats, earthworms (*Eisenia fetida* (Savigny, 1826)) and TiO$_2$ nanoparticles.**

| Treatment | TiO$_2$-nanoparticles (mg kg$^{-1}$ dry soil) |
| --- | --- |
| Unamended | 0 |
| | 150 |
| | 300 |
| Oats: 1.75 g tyndallized Quaker® oat kg$^{-1}$ | 0 |
| | 150 |
| | 300 |
| Earthworms: 10 earthworms with fully developed clitellum and average fresh mass of 0.5 to 500 g dry soil | 0 |
| | 150 |
| | 300 |
| Oats plus earthworms | 0 |
| | 150 |
| | 300 |

Soil was sampled 30 times at random from the 0–15 cm top-layer of three plots of approximately 0.6 ha. The soil from each plot was pooled so that three soil samples were obtained. These three soil replicates were maintained separately in the laboratory study and characterized. Details of the techniques used to characterize the soil are given in section 2.8. The loamy soil with pH 7.8 had an electrolytic conductivity (EC) 0.6 dS m$^{-1}$, water holding capacity (WHC) 547 g kg$^{-1}$, organic C content 11.7 g C kg$^{-1}$ soil and a total N content 0.94 g N kg$^{-1}$ soil.

## Vessel preparation and experimental design

The soil from each plot was passed separately through a five mm sieve. Samples of 500 g dry soil of each plot was adjusted to 40% WHC by adding distilled H$_2$O. The soil samples were placed in amber glass jars and pre-incubated at 25 ± 2 °C in drums containing a beaker with 100 ml 1M sodium hydroxide to trap CO$_2$ evolved, and a beaker with 100 ml distilled H$_2$O to avoid desiccation of the soil for 7 days (*Bundy & Bremner, 1972*).

After 7 days, 12 different treatments were applied to the soil and details can be found in Table 2. A third of the soil samples were adjusted to 60% WHC by adding distilled H$_2$O, a third were amended with an aqueous solution of TiO$_2$-NP to a concentration of 150 mg TiO$_2$-NP kg$^{-1}$ soil and the remaining third to 300 mg TiO$_2$-NP kg$^{-1}$ dry soil. The unamended soil, and soil amended with 150 or 300 mg TiO$_2$-NP kg$^{-1}$ dry soil were then added with oats, earthworms or oats plus earthworms. All samples were homogenized manually with a spatula before the earthworms were applied.

Each amber glass jar was sealed with mosquito net to avoid anaerobic conditions and pests, and to prevent the escape of the earthworms. The glass jars were placed in a plant growth chamber with average temperature of 20 ± 2 °C and a photoperiod of 12 h light and 12 h dark. After 15, 30, 60 and 90 days, a flask from each treatment ($n = 12$) and replicate sample ($n = 3$) was removed and analyzed for pH, EC, WHC, NO$_2^-$, NO$_3^-$, and organic C content.

## Sampling of adult, juvenile and cocoons worms

After 15, 30, 60 and 90 days, the soil was removed from the glass jar and the adult worms, juveniles and cocoons were separated by hand and counted.

## DNA extraction and PCR amplification of 16S rRNA gene

Metagenomic DNA was extracted with three different techniques each from a 0.5 g soil sub-sample and pooled (*De León-Lorenzana et al., 2017*). As such, 1.5 g soil from each plot ($n = 3$), treatments ($n = 12$) and sampling day ($n = 5$) was extracted for metagenomic DNA. Overall, 270 g soil was extracted and 180 DNA samples were obtained.

The V3-V4 hypervariable regions of 16S rRNA genes were amplified with DNA polymerase (Thermo Fisher Scientific, Hudson, NH, USA) in a final 25 µl reaction volume. Amplification was done with the PCR Thermal Cycler Multigene Optimax (Labnet, Edison, NJ, USA). The following thermal cycling scheme was used: initial denaturation at 94 °C for 10 min; 25 cycles of denaturation at 94 °C for 45 s, annealing at 53 °C for 45 s, and extension at 72 °C for 1 min; followed by a final extension at 72 °C for 10 min, using the forward primer 341F (5′-CCTACGGGIGGCWGCAG-3′) and the reverse primer 805R (5′-GACTACHVGGGTATCTAATCC-3′) containing the Illumina platform adapters and eight bp barcodes. All samples were amplified in triplicate, pooled in equal volumes and sequenced by Macrogen Inc. (DNA Sequencing Service, Seoul, Korea) with paired-end Illumina MiSeq sequencing system (Illumina, San Diego, CA, USA).

## Analysis of the soil microbial community

Sequences were analyzed with the QIIME version 1.9.1 (*Caporaso et al., 2010*). Sequences were filtered for quality score (minimum 30% Phred quality) and sequences with mismatches in the barcode or in primers were removed from the datasets. The number of operational taxonomic units (OTUs) was determined at the 97% similarity level using the UCLUST algorithm (*Edgar, 2010*). Taxonomic assignation of 16S rRNA was done with the Greengenes core-set-aligned with UCLUST (http://greengenes.lbl.gov/). A total of 4,526 OTUs were selected at random per sample for subsequent analysis. All sequences were deposited in NCBI Sequence Read Archive under the BioProject accession number PRJNA453453: Soil with NP of TiO$_2$ Raw sequence reads. PICRUSt v. 1.1.1 software (http://picrust.github.io (*Langille et al., 2013*)) was used to predict the functionality of bacterial communities using the Kyoto encyclopedia of genes and genomes pathways with OTUs assigned at the 97% identity using QIIME version 1.9.1 with a closed reference strategy against Greengenes database 13_5 (http://greengenes.lbl.gov/).

## Characterization of the soil samples

The EC was measured by saturating 50 g soil to 100% WHC with distilled water, left to stand overnight at 4 °C and centrifuged. The EC was determined in the supernatant with a microprocessor HI 933300 (HANNA Instruments, Woonsocket, RI, USA). The pH was determined in 1:2.5 soil-H$_2$O suspension using a calibrated Ultra Basic UB-10 pH/mV meter (Denver Instrument, New York, NY, USA) fitted with a glass electrode (#3007281 pH/ATC, ThermoFisher Scientific, Waltham, MA, USA). The WHC was measured in 50 g

water saturated soil samples placed in a funnel, covered with aluminum foil to avoid water loss and allowed to stand overnight to drain freely. The amount of water retained in the soil was defined as the WHC (*De León-Lorenzana et al., 2017*). The extraction and quantification of $NO_2^-$ and $NO_3^-$ was done as described by *Ramírez-Villanueva et al. (2015)* and the total C content as described by *Navarro-Noya et al. (2013)*.

### Statistical analysis

All analyses were done in *R Development Core Team (2014)*. A parametric test (ANOVA, Type III *F*-test) was used to determine the effect of the application of NP, earthworms and oats on soil characteristics, and a non-parametric test to determine the effect of application of NP and oats on the characteristics of the earthworms. The t2way test of the WRS2 package (A collection of robust statistical methods) was used (*Mair & Wilcox, 2017*). Abundance of the different bacterial taxonomic levels was explored separately with a principal component analysis (PCA) and constrained analysis of principal coordinates (CAP) was used to explore the effect of treatment on the bacterial groups and soil characteristics. *Boslaugh (2013)* stated: "a *PCA is based on an orthogonal decomposition of an input matrix to yield an output matrix that consists of a set of orthogonal components that maximize the amount of variation in the variables from the input matrix.*" In this study, a PCA was used to visualize patterns and variations in the bacterial populations of the different soil samples, while CAP was used for the same reason but included two datasets in the analysis, that is, bacterial populations and soil characteristics. These analyses were done with the vegan package (*Oksanen et al., 2017*). Heatmaps were constructed with the pheatmap package (*Kolde, 2015*).

## RESULTS AND DISCUSSION

### Effect of soil amendments on soil characteristics

The pH of the arable soil was 7.8 at the onset of the experiment. Application of oats, earthworms or the combination of both reduced pH significantly compared to the unamended soil after 90 days ($p < 0.01$) (Figs. 1A and 1B). The EC of the arable soil was 1.08 dS m$^{-1}$ at day 0 (Figs. 1C and 1D). Application of oats, earthworms or the combination of both increased EC significantly compared to the unamended soil after 90 days, with the largest effect found when both earthworms and oats were added ($p < 0.01$). After 90 days, the total C and WHC in soil was not affected significantly by the treatments applied to soil (Figs. 1E–1H). Application of oats, earthworms or the combination of both increased the concentration of $NO_2^-$ and $NO_3^-$ significantly compared to the unamended soil after 90 days ($p < 0.01$) (Figs. 1I–1L). Application of the NP had no significant effect on pH, EC, total C, WHC, or the amount of $NO_2^-$ and $NO_3^-$ after 90 days (Fig. 1).

The pH and EC are defined by different soil characteristics, such as organic material (*Bot & Benites, 2005*). Application of organic material and its subsequent mineralization alters pH (https://www.nrcs.usda.gov/Internet/FSE_DOCUMENTS/nrcs142p2_053293.pdf). Earthworms feed on soil organic matter and the microorganisms in their gut accelerate its degradation (*Thomason et al., 2017*). The mineralization of the organic material changes pH and the change is more accentuated when earthworms are more

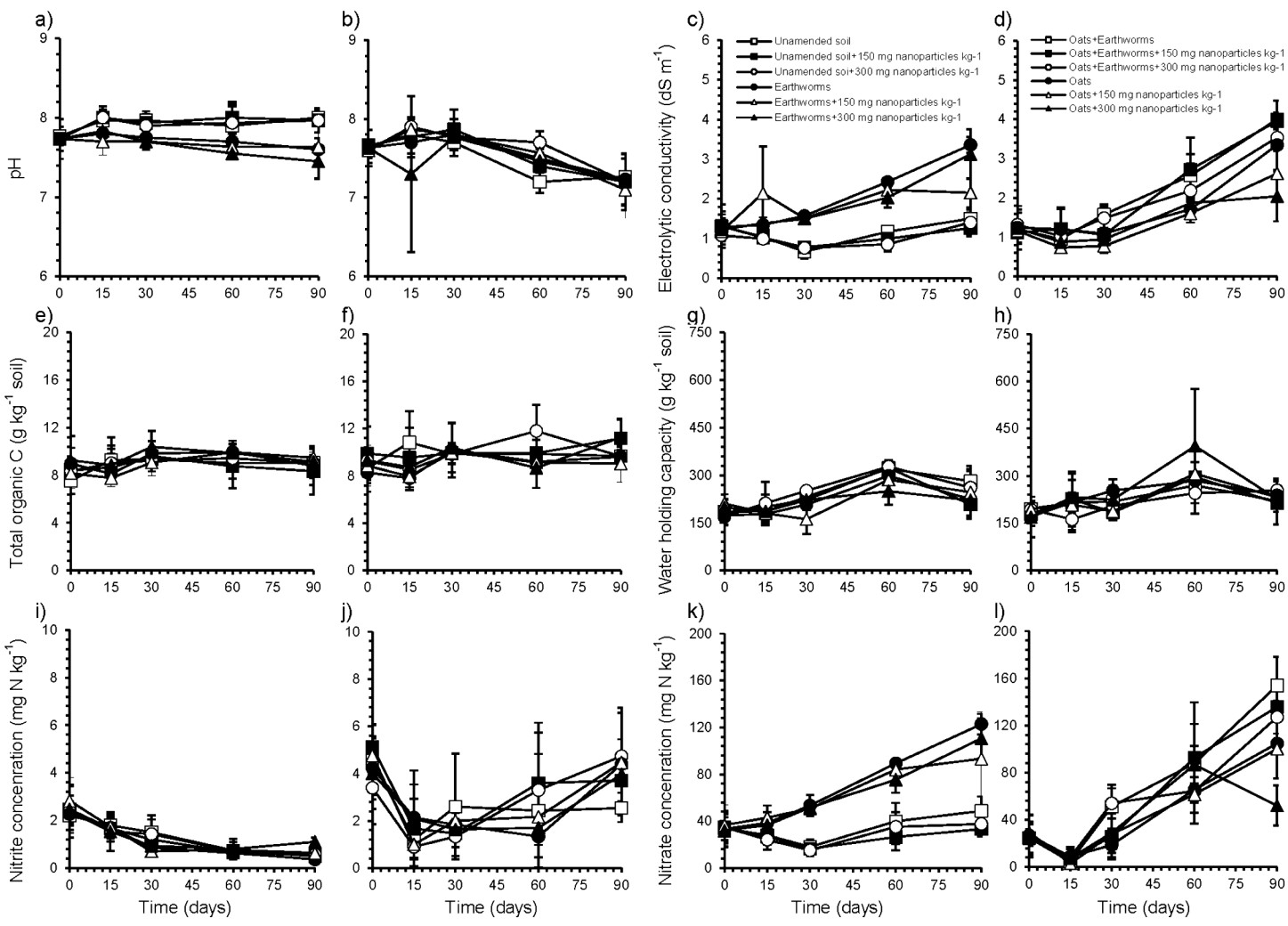

**Figure 1 Effect of soil amendments on soil characteristics.** (A) pH, (C) electrolytic conductivity, (E) organic carbon content, (G) water holding capacity, (I) concentration of nitrite ($NO_2^-$), (K) concentration of nitrate ($NO_3^-$) in **unamended soil** contaminated with 0, 150 or 300 mg nanoparticles $kg^{-1}$, and **soil applied with earthworms** (*Eisenia fetida* (Savigny, 1826)) and contaminated with 0, 150 or 300 mg nanoparticles $kg^{-1}$ after 0, 15, 30, 60 and 90 days. (B) pH, (D) electrolytic conductivity, (F) organic carbon content, (H) water holding capacity, (J) concentration of nitrite ($NO_2^-$), (L) concentration of nitrate ($NO_3^-$) in the **soil amended with oats** and contaminated with 0, 150 or 300 mg nanoparticles $kg^{-1}$, and **amended with oats plus earthworms** and contaminated with 0, 150 or 300 mg nanoparticles $kg^{-1}$ after 0, 15, 30, 60 and 90 days.

active in soil. The organic N mineralized is liberated as $NH_4^+$ and oxidized by nitrifiers to $NO_2^-$ and subsequently $NO_3^-$ (*Signor & Cerri, 2013*). This oxidation generates a proton, which decreases pH. Apart from releasing $NH_4^+$, mineralization of organic material will release other cations and anions that will increase the EC as found in this study.

The untreated unamended arable soil was N depleted as the amount of $NO_3^-$ decreased until day 30. The immobilized N was then released, but the amount of N mineralized was low after 90 days, that is, <15 mg N $kg^{-1}$ soil. In conventional agricultural practices with maize monoculture most of the crop residue is removed and little inorganic N fertilizer is applied. The crop residue that is left has a high C-to-N ratio (57 to 1 (*USDA NRCS, 2018*)) that will stimulate N immobilization temporarily. The earthworm

activity accelerated organic material mineralization so no N immobilization was detected in the earthworm amended soil and the amount of $NO_3^-$ -N increased with 74 mg $NO_3^-$ -N. The application of organic material increases the amount of $NO_2^-$ and $NO_3^-$ in soil if its C-to-N ratio is low, that is, <20 (*Akiyama & Tsuruta, 2003*). If the organic material applied to soil contains mostly recalcitrant compounds, for example, lignin, and/or its C-to-N ratio is high, then little or no mineral N will be released in soil and N immobilization can occur temporally (*Dendooven, 1990*). The oats applied were mostly easily degradable and had a C-to-N ratio 20.2 (total C was 449 g $kg^{-1}$ and total N 22.2 g $kg^{-1}$). Although the C-to-N ratio was low some N immobilization occurred in the first 15 days of the incubation. The immobilized N was mineralized again and already at day 30 the amount of $NO_3^-$ increased. After 90 days, the $NO_3^-$ concentrations had increased with 56 mg $NO_3^-$-N $kg^{-1}$ soil. Consequently, combining oats plus earthworms further increased the amount of mineral-N, that is, 105 mg N $kg^{-1}$ soil.

The effect of different kinds of NP on C and N dynamics has been studied before. For instance, *Simonin et al. (2015)* found that $TiO_2$-NP did not affect microbial activity in five soils studied, but in a silty clay soil with high organic material C mineralization was inhibited. They suggested that a possible effect of $TiO_2$-NP did not depend on soil texture, but was controlled by pH and soil organic matter content. In this study, $TiO_2$-NP had no significant effect on the N mineralized or on the nitrification process as evidenced by the concentration of $NO_2^-$. This does not mean, however, that microorganisms involved in the ammonification or nitrification process were not affected, but the release of inorganic N was not inhibited (*McKee & Filser, 2016*).

## Effect of soil amendments on earthworms

The number of adult earthworms remained constant in the oats-amended soil, but decreased when no feed, that is, oats, was given to the earthworms after 90 days and reproduction was almost zero (Fig. 2A). The number of juvenile earthworms increased sharply and near linearly after day 30 when oats were applied to soil but was not significantly affected by the application of NP (Fig. 2B). The number of cocoons remained constant when no feed was applied to soil, but showed a maximum when oats were applied to soil at day 60 (Fig. 2C). The number of shells of *E. fetida* cocoons remained constant when no feed was applied to soil, but showed a maximum when oats were applied to soil at day 60, except when no NP were added to soil (Fig. 2D). When the soil is not supplemented with organic material, earthworms lack food and they die as happened in the unamended soil toward the end of the incubation (*Weyers & Spokas, 2011*). Application of oats provided them with food so that the number of cocoons and juveniles was higher than in the unamended soil. The number of adult and juvenile earthworms, cocoons and cocoon shells was not affected significantly by the application of NP in soil amended with oats or left unamended (Table S2).

*Cañas et al. (2011)* placed *E. fetida* on filter paper with $TiO_2$-NP and found that the survival of earthworms was reduced after 14 days. However, some studies even reported no effect of $TiO_2$-NP on the survival of earthworms in soil even at application rates of one g $kg^{-1}$ soil (*Bigorgne et al., 2012*; *Hu et al., 2010*; *McShane et al., 2012*). For instance,

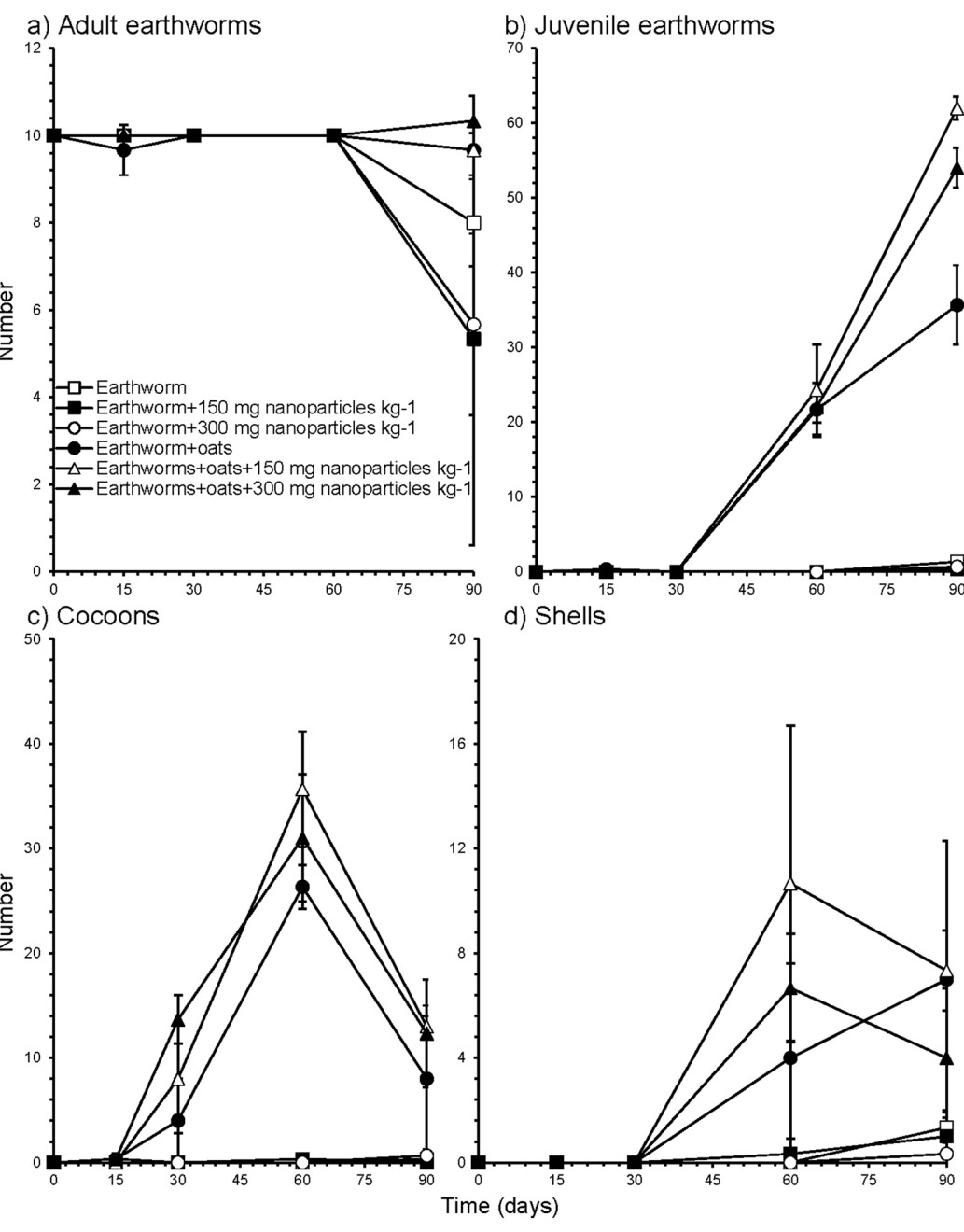

**Figure 2 Effect of soil amendments on earthworms.** The number of (A) adult (B) juvenile, (C) cocoons and (D) shells of *Eisenia fetida* (Savigny, 1826) in soil applied with 0, 150 or 300 mg nanoparticles kg$^{-1}$ amended with **earthworms** or **earthworms plus oats** at the onset of the experiment and after 15, 30, 60 and 90 days.                                               

*Johnson et al. (2011)* reported that coelomocytes of *E. veneta* (Rosa), the major component of their immune system, was not affected by direct exposure to TiO$_2$ engineered NP. *Eisenia andrei* (*Bouché, 1972*), however, avoided TiO$_2$-NP in concentrations between one and 10 g kg$^{-1}$ (*Schlich et al., 2013*). Some studies reported even a stimulating effect of TiO$_2$-NP on earthworm survival and the number of cocoons (*Whitfield Åslund et al., 2011*), which was not confirmed in this study.

Although no effect of TiO$_2$-NP on *E. fetida* was detected in this study, it is possible that it might induce oxidative stress, inhibition of cellulose activity and cause damage to DNA of the earthworm at concentrations higher than 1.0 g kg$^{-1}$ soil as reported by *Hu et al. (2010)*. A possible negative effect of TiO$_2$-NP depends on their bioavailability. The TiO$_2$-NP might be fixed on the soil organic matter and clay particles (*Shi, Tang & Wang, 2017*), thus preventing them from interacting with microorganisms or macroorganisms (*Bigorgne et al., 2011*). However, as the earthworms borough through the soil the TiO$_2$-NP fixed on soil particles and/or organic material pass through their gut. The TiO$_2$-NP might accumulate in the earthworm for a while or after the organic material is digested and liberated again (*Bigorgne et al., 2012*). The soil used in this study had a high clay content 540 g kg$^{-1}$, which might have reduced the bioavailability of TiO$_2$-NP.

## Bacterial community structure in the unamended soil

Thirty bacterial phyla were detected in the unamended soil with Proteobacteria the most abundant (37.76 ± 2.50%) (mostly Alphaproteobacteria (24.11 ± 2.69%) and Gammaproteobacteria (8.24 ± 1.56%)), followed by Acidobacteria (19.65 ± 2.59%) (mostly Acidobacteria-6 (9.62 ± 1.14%)) and Actinobacteria (14.07 ± 1.85%). The relative abundance of the most abundant bacterial phyla showed little changes over time (Fig. S2A). The relative abundance of Armatimonadetes, Bacteroidetes and Verrucomicrobia was larger at the end of the experiment than at the beginning, and that of BRC1, Nitrospirae and OD1 showed an opposite trend (Figs. 3A and 3C).

*Novosphingobium* (relative abundance 8.36 ± 2.49%) was the most abundant genus, followed by *Kaistobacter* (relative abundance 2.91 ± 0.54%) both belonging to the Sphingomonadales, which was the most abundant bacterial order (12.48 ± 3.28%). The relative abundance of the most abundant bacterial genera showed only small changes over time except in the soil applied with oats plus earthworms at day 60 and 90 (Fig. S3). The PCA separated the bacterial community at the different sampling days from each other as the relative abundance of *Sphingobacterium* and *Streptomyces* increased over time, while that of *Novosphingobium* and *Kaistobacter* decreased (Figs. 3B and 3D). The functionality of the bacterial community in the unamended soil showed also a change with incubation time, although not so clear as when considering the bacterial community structure (Figs. 3F and 3H).

The microbial community structure is defined by soil conditions, for example, organic material, water content, clay content, pH or salt content (*Docherty et al., 2015*). Often, the bacterial community in arable soil is dominated by Proteobacteria, Acidobacteria or Actinobacteria as in this study (*Asadishad et al., 2018*). Organic material is of paramount importance for most microorganisms (heterotrophs) as it serves as C substrate, and/or to provide energy (*Roller & Schmidt, 2015*). Incubating the soil for 90 days reduced the organic material as the easily decomposable part was mineralized leaving mostly resistant or recalcitrant organic material. As such, copiotrophs (microorganisms enriched by organic material) should be favored at the onset of the experiment and oligotrophs (favored by nutrient poor environments) toward the end of the incubation (*Fierer et al., 2012*; *Yao et al., 2017*). pH is the factor that often controls the bacterial community and

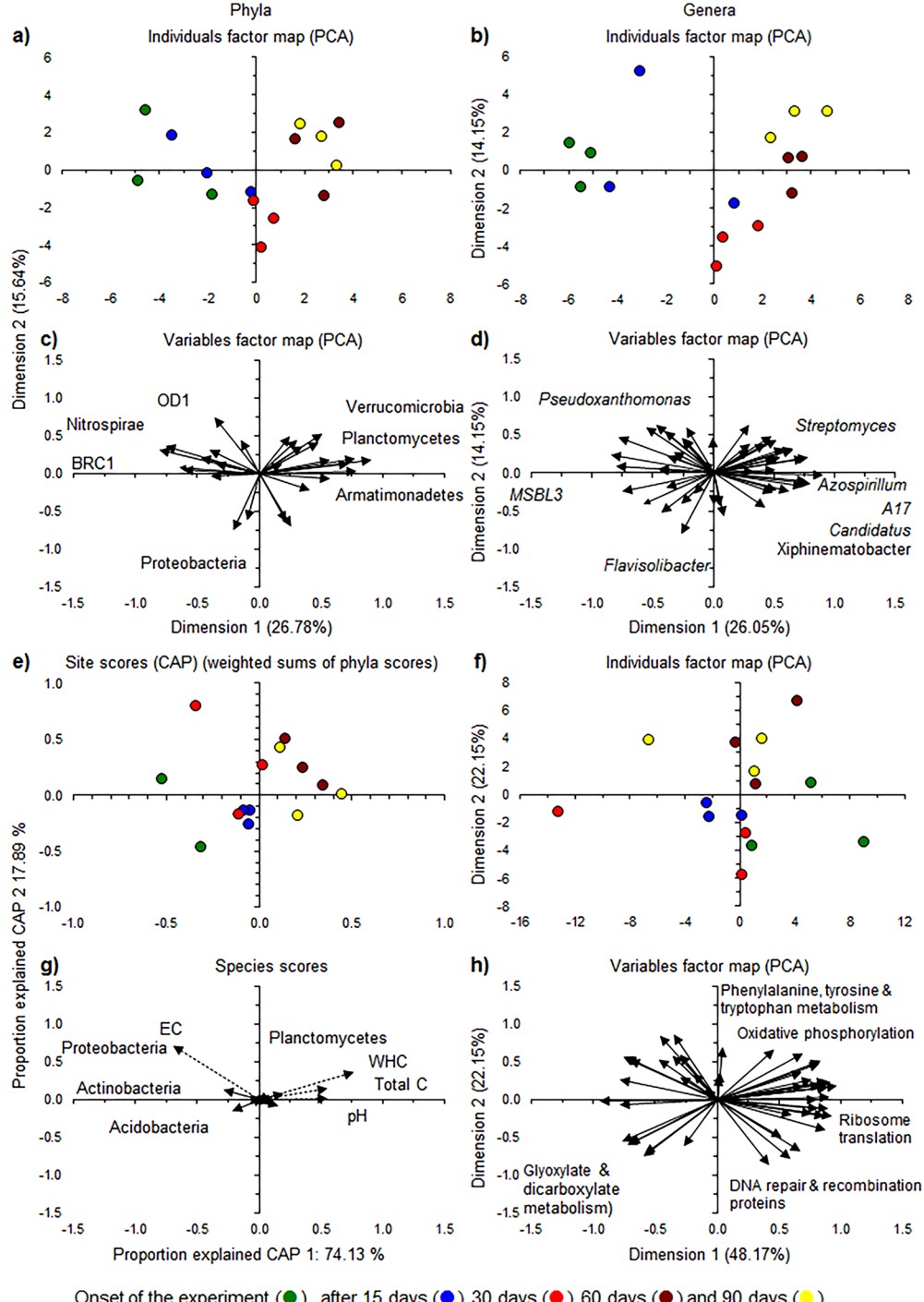

Onset of the experiment (●), after 15 days (●), 30 days (●), 60 days (●) and 90 days (●)

**Figure 3 Relative abundance of the bacterial phyla and genera.** (A) Individuals factor map and (C) Variables factor map of the Principal component analysis (PCA) with the relative abundance of the most abundant bacterial phyla; (B) Individuals factor map and (D) Variables factor map of PCA of the 50 most abundant bacterial genera; (E) Site scores and (G) Species scores of the constrained analysis of principal coordinates (CAP) with the relative abundance of bacterial phyla and selected soil character-istics; (F) Individuals factor map and (H) Variables factor map of the PCA with the most important functions of the bacterial communities using the Kyoto encyclopedia of genes and genomes (KEGG) pathways.                                                           

even small changes in pH might alter the relative abundance of several bacterial groups (*Bang-Andreasen et al., 2017*; *Zhang et al., 2017a*). Within 90 days, the bacterial community was different from that at the onset, presumably as a combined effect of an increase in pH and a decrease in easily decomposable soil organic material. In this experiment, the Actinobacteria, Armatimonadetes, Cyanobacteria, Firmicutes and Planctomycetes behaved as oligotrophs, that is, their relative abundance increased within 90 days, and Acidobacteria, BRC1, FBP, Nitrospirae and OD1 as copiotrophs, that is, their relative abundance decreased within 90 days. In this study, Actinobacteria are considered copiotrophs generally (*Peiffer et al., 2013*), but they behaved as oligotrophs and Acidobacteria are considered oligotrophs normally (*Fierer et al., 2012*), but they behaved as copiotrophs. It is clear that soil conditions will define how bacteria respond to a decrease in soil organic matter and changes in pH. As such, the ecological coherence of high bacterial ranks as suggested by *Philippot et al. (2010)* through genome analysis can be questioned.

## Effect of soil amendments on the bacterial community structure

The application of oats, earthworms or a combination of both affected the relative abundance of different bacterial groups (Fig. S4). Eight different categories of bacterial groups could be distinguished based on how they responded to the application of the organic material, earthworms or the combination of both (Fig. S4; Table 3).

The application of oats, earthworms and oats plus earthworms altered the bacterial community structure clearly as visualized by a PCA (Figs. 4A–4F). The relative abundance of Bacteroidetes and Proteobacteria increased in the oats-amended soil, that of Actinobacteria and Verrucomicrobia increased in the earthworm-amended soil and Bacteroidetes, Proteobacteria and Verrucomicrobia when both were combined compared to the unamended soil. The effect was even more accentuated when the most dominant genera were considered (Figs. 5A–5F). The relative abundance of bacterial genera, such as *Kaistia*, *Pedobacter* and *Zobellella*, increased in the oats-amended soil (Figs. 5A and 5D), *Gemmata*, *Geodermatophilus* and *Vibrio* in the earthworm-amended soil (Figs. 5B and 5E), and *Agrobacterium*, *Anaerovirgula* and *Luteolibacter*, in the oats plus earthworms amended soil over time compared to the unamended soil (Figs. 5C and 5F). The CAP analysis showed also an effect of incubation time on the bacterial community structure most accentuated in the oats plus earthworms amended soil (Figs. 6A, 6B, 6D and 6E).

The PCA analysis with the different metabolic functionalities of the bacterial groups showed also an effect of incubation time in the oats plus earthworms amended soil and to a lesser extent in the oats-amended soil compared to the unamended or earthworm-amended soil (Figs. 6C and 6F). Metabolic functions, such as alanine, aspartate and glutamate metabolism, oxidative phosphorylation, ribosome translation and valine, leucine and isoleucine biosynthesis decreased in the earthworm plus oats and oats-amended soil compared to the unamended or earthworm-amended soil over time while that of other ion-coupled transporters, transcription factors and membrane transport increased.

Application of organic material to soil is known to affect the bacterial community structure (*Fernandez et al., 2016*). Some bacteria that degrade the organic material are

**Table 3  The different categories of the bacterial phyla and most dominant genera as defined by how they respond to the application of oats, earthworms (*Eisenia fetida* (Savigny, 1826)) or the combination of both.**

1. Relative abundance of the bacterial group was **higher** when oats, earthworms and the combination of both were applied to soil compared to the unamended soil.

| Bacteroidetes, Firmicutes, Tenericutes and Verrucomicrobia | *Acidovorax, Agrobacterium, Agromyces, Anaerovirgula, Chitinophaga, Citrobacter Flavobacterium, MSBL3* (Verrucomicrobiaceae), *Pedobacter, Pseudomonas, Sphingobacterium, Hyphomicrobium, Zobellella* |
|---|---|

2. Relative abundance of the bacterial group was **lower** when oats, earthworms and the combination of both were applied to soil compared to the unamended soil.

| Acidobacteria, Fibrobacteres, Gemmatimonadetes, Nitrospirae and OP3 | *Azospirillum, Candidatus* Xiphinematobacter, *Devosia, Kaistobacter, Novosphingobium, Planctomyces, Rubrobacter, Sphingopyxis* |
|---|---|

3. Relative abundance of the bacterial group was **higher** when oats were applied to soil compared to the unamended soil, but **lower** when earthworms and the combination of both were added.

| Actinobacteria, Armatimonadetes, BRC1, Chlorobi, Chloroflexi, Cyanobacteria, Planctomycetes, TM7, (Thermi) | *Amaricoccus, Balneimonas, Bosea, Cupriavidus* |
|---|---|

4. Relative abundance of the bacterial group was **lower** when oats were applied to soil compared to the unamended soil, but **higher** when earthworms and the combination of oats and earthworms were added.

| NKB19, Proteobacteria | *Adhaeribacter, Arthrospira, Luteolibacter* |
|---|---|

5. Relative abundance of the bacterial group was **higher** when earthworms were applied to soil compared to the unamended soil, but **lower** when oats and the combination of oats and earthworms were added.

| OD1, WS2 | *Aeromicrobium, Gemmata, Geodermatophilus, Mycobacterium, Pseudonocardia, Steroidobacter, Streptomyces, Vibrio* |
|---|---|

6. Relative abundance of the bacterial group was **lower** when earthworms were applied to soil compared to the unamended soil, but **higher** when oats and the combination of oats and earthworms added.

| Chlamydiae | *Dysgonomonas, Kaistia, Rhodobacter, Thermomonas* |
|---|---|

enriched, that is, copiotrophs, and others not, that is, oligotrophs (*Koch, 2001*). Which organisms are enriched depends on the organic material applied and the soil characteristics. Bacteroidetes and Proteobacteria are generally enriched by application of organic material (*Fierer et al., 2012*) as in this study. The relative abundance of most bacterial phyla decreased, for example, Nitrospirae, and some others have been described previously as oligotrophs, for example, Acidobacteria (*Elliott et al., 2015*). However, Actinobacteria are considered copiotrophs (*Fierer et al., 2012*), but they were not enriched when oats were applied to soil. Again, the ecological coherence of high bacterial taxa (from genus to phylum) could not be confirmed as not all members of a bacterial group responded in the same way to the application of the organic material.

Earthworm activity will accelerate the degradation of organic material and this will change soil characteristics, such as soil structure and pH (*Bernard et al., 2012*). These changes controlled by the initial soil conditions, food and earthworm activity might alter the bacterial community structure (*Lin et al., 2018*). In this study, the bacterial community structure was different between the unamended soil and the earthworm-amended soil already after 30 days. The activity of the earthworms enriched Actinobacteria (mostly *Agromyces*), TM6 and Verrucomicrobia (mostly MSLB3). Members of *Alicyclobacillus*, *Anaerovirgula* and *Geobacillus* (Firmicutes) were also enriched in the earthworm-amended soil and some of their members are characterized by cellulolytic activity. *Alicyclobacillus* consists of a group of thermo-acidophilic, strictly aerobic, heterotrophic

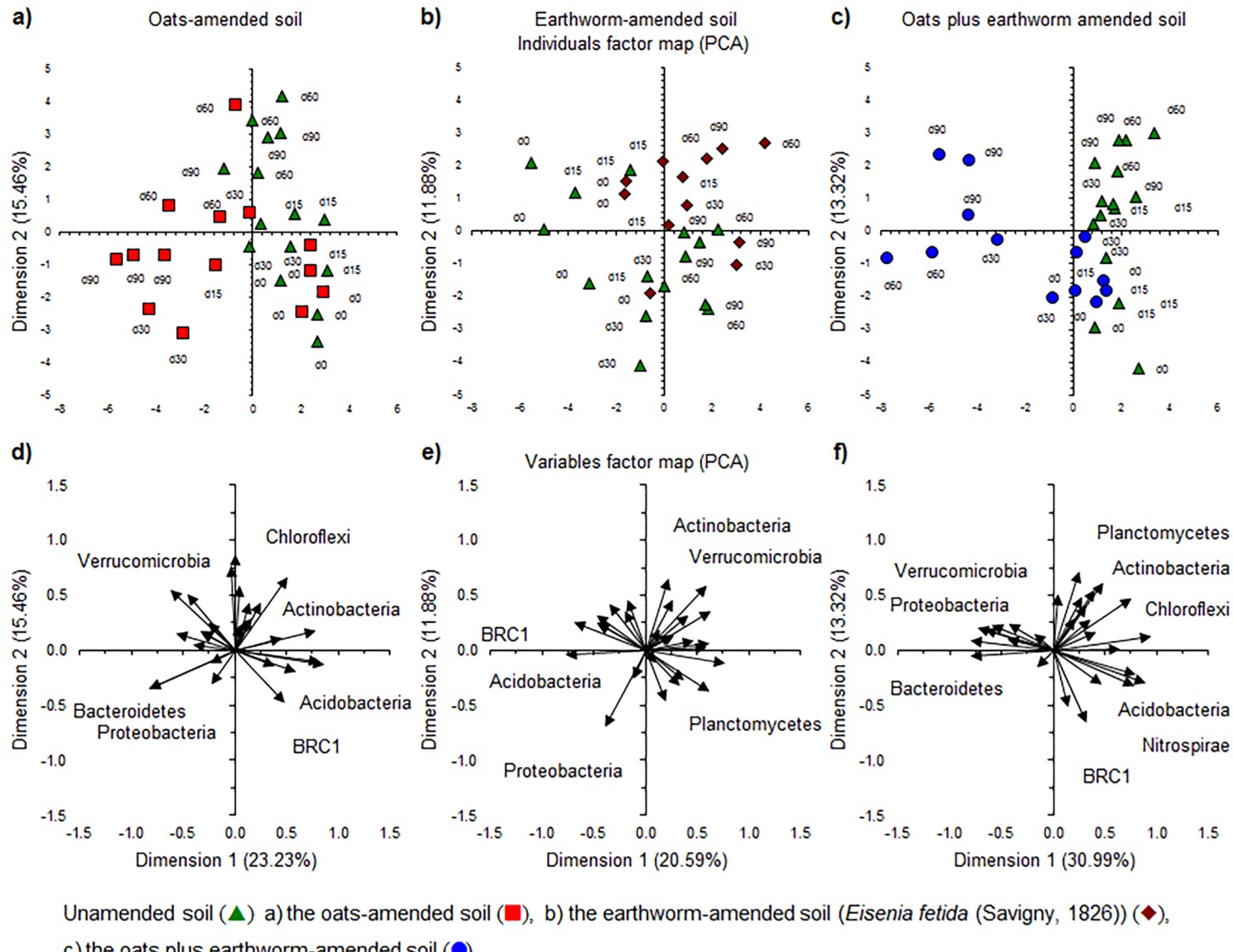

Unamended soil (▲) a) the oats-amended soil (■), b) the earthworm-amended soil (*Eisenia fetida* (Savigny, 1826)) (◆),

c) the oats plus earthworm-amended soil (●)

**Figure 4 Principal component analysis (PCA) with the relative abundance of the bacterial phyla.** Individuals factor maps and Variables factor maps of the unamended soil vs. oats-amended soil, (A) and (D); unamended soil vs the earthworm-amended soil (*Eisenia fetida* (Savigny, 1826)), (B) and (E); and unamended soil vs. earthworm-amended soil, (C) and (F); after 0 days (d0), 15 days (d15), 30 days (30d), 60 days (60d) or 90 days (90d) of an aerobic incubation.                              

and spore-forming bacteria (*Ciuffreda et al., 2015*) with some members with cellulolytic capacity, for example, *Alicyclobacillus cellulosilyticus* (*Kusube et al., 2014*). *Anaerovirgula* is an alkaliphilic, cellulolytic obligate anaerobic bacterium (*Porsch et al., 2015*), while *Geobacillus* produces thermostable hemicellulose hydrolytic enzymes (*De Maayer et al., 2014*). *Lin et al. (2018)* contaminated soil with atrazine and applied two earthworms, *E. fetida* and *Amynthas robustus* (E. Perrier). The former is an epigeic species or living in the top soil, and the latter is an endogeic species that burrows through all soil layers. They reported that members of *Rhodoplanes* and *Kaistobacter* participated in the mineralization of atrazine and were enriched by both earthworm species. *Amynthas*
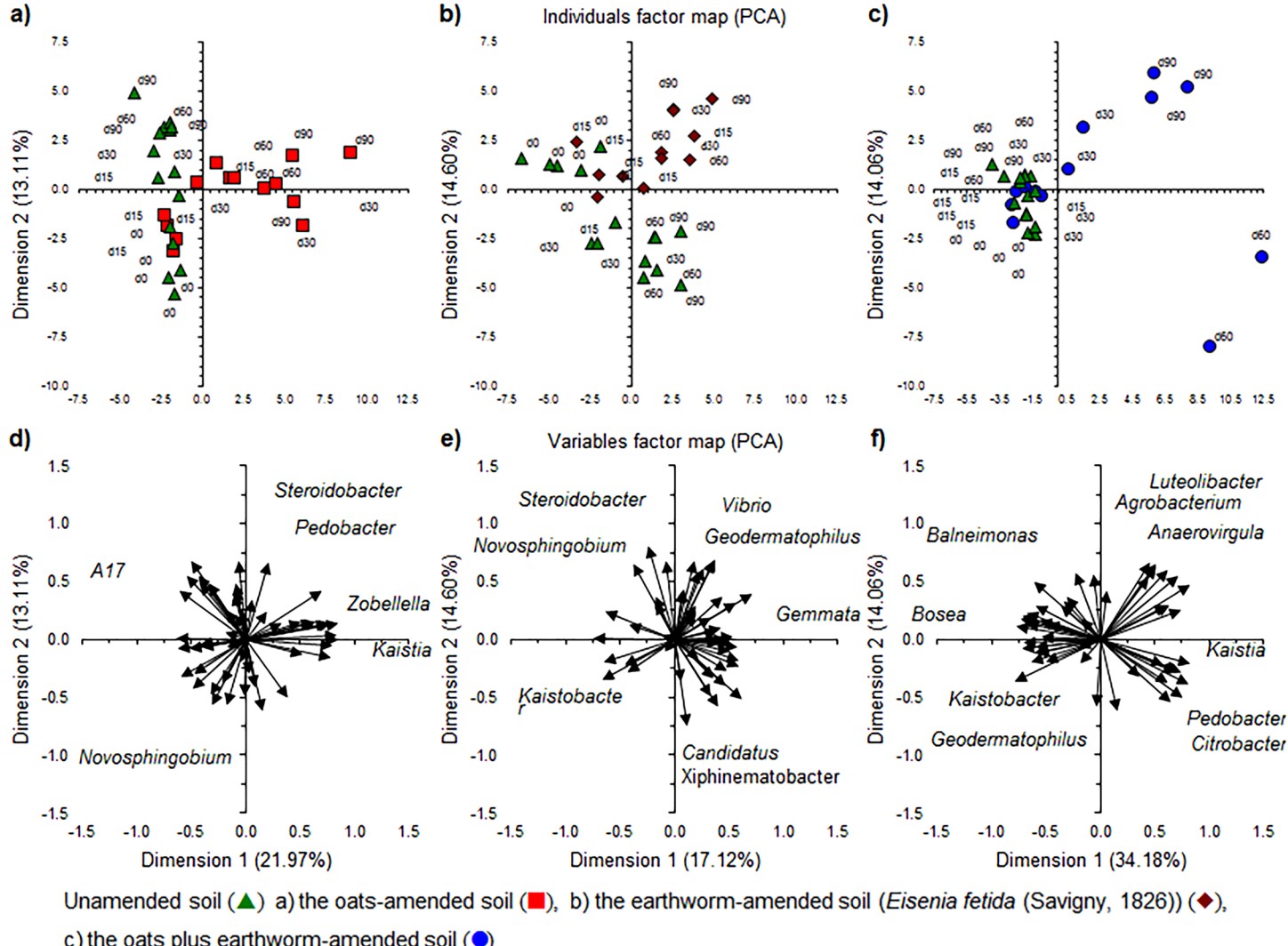

Unamended soil (▲) a) the oats-amended soil (■), b) the earthworm-amended soil (*Eisenia fetida* (Savigny, 1826)) (◆), c) the oats plus earthworm-amended soil (●)

**Figure 5  Principal component analysis (PCA) with the relative abundance of the 50 most abundant bacterial genera.** Individuals factor maps and Variables factor maps of the unamended soil vs. oats-amended soil, (A) and (D); unamended soil vs. the earthworm-amended soil (*Eisenia fetida* (Savigny, 1826)), (B) and (E); and unamended soil vs. earthworm-amended soil, (C) and (F); after 0 days (d0), 15 days (d15), 30 days (30d), 60 days (60d) or 90 days (90d) of an aerobic incubation.

*robustus* increased the relative abundance of *Cupriavidus* and *Pseudomonas*, while *Flavobacterium* was enriched by *E. fetida*. In this study, *Pseudomonas* and *Flavobacterium* were enriched by *E. fetida*, but the relative abundance of *Cupriavidus* and *Kaistobacter* decreased.

The combined application of oats plus earthworms enriched members of the Bacteroidetes, Proteobacteria, TM6 and Verrucomicrobia, while reducing most other bacterial phyla. First, not all bacterial groups that belonged to the phyla that were enriched by the application of oats plus earthworms responded in the same way. Second, the effect of earthworms or oats on the relative abundance of a bacterial group was not always the same as when earthworms plus oats were applied to soil. For instance, *Agromyces* was enriched by the application of oats plus earthworms, but not *Geodermatophilus* both

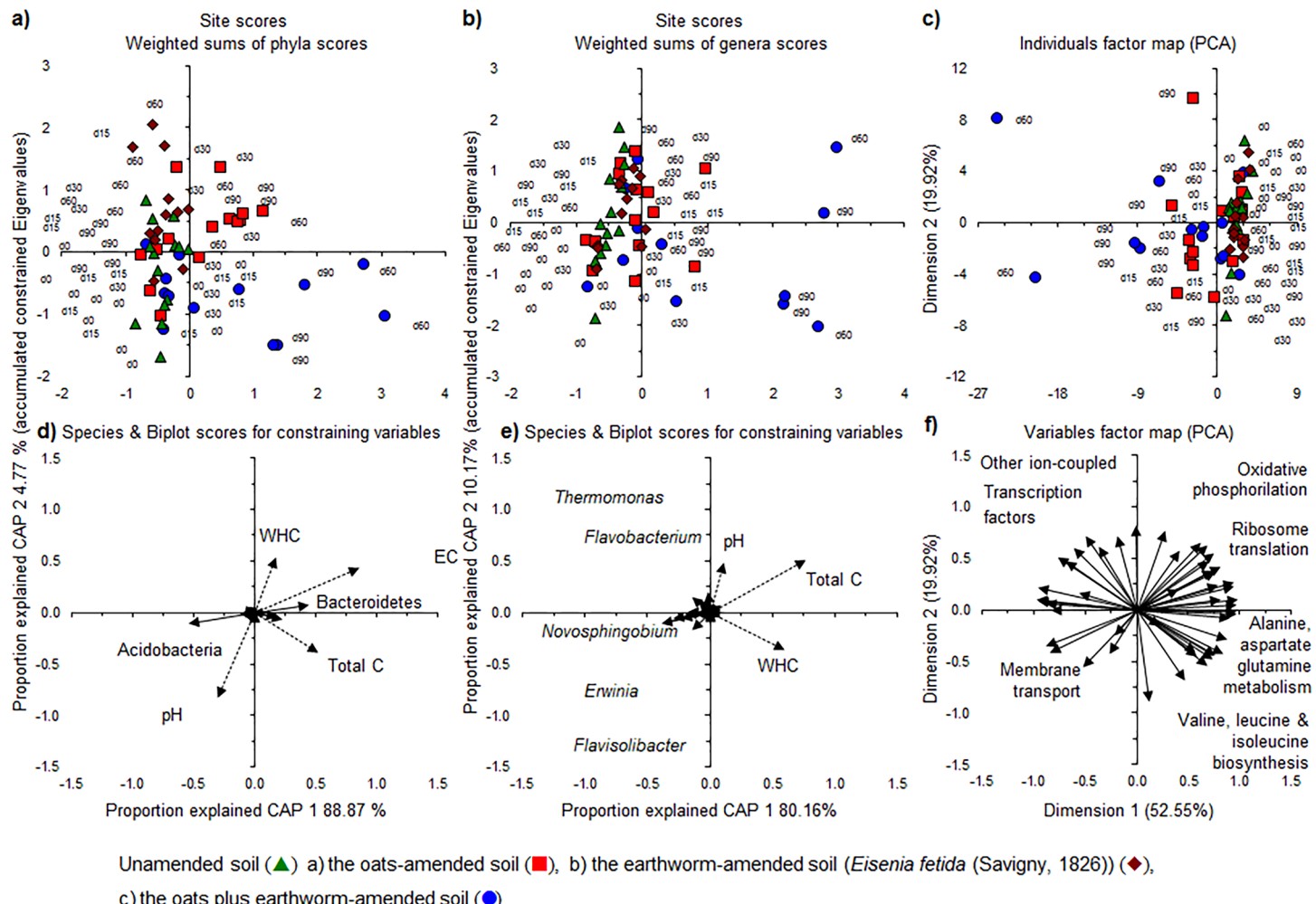

**Figure 6 A constrained analysis of principal coordinates (CAP) and Principal Components of bacterial phyla and the most abundant genera, including the soil characteristics and metabolic functionalities.** Site scores and Species & Biplot scores for constraining variables, (A) and (D), of the bacterial phyla; Site scores and Species & Biplot scores for constraining variables, (B) and (E), of the 50 most abundant genera; and Individuals factor map and Variables factor map, (C) and (F), of a principal component analysis (PCA) with metabolic functionalities; after 0 days (d0), 15 days (d15), 30 days (30d), 60 days (60d) or 90 days (90d) of an aerobic incubation.  

belonging to the Actinomycetales. It has to be remembered that the bacterial groups compete and affect each other, and that interaction will be controlled by soil conditions. Consequently, some members of a bacterial group might be inhibited by application of oats or earthworms, but not when both treatments were combined, for example, TM6 and *Erwinia*.

## Effect of nanoparticles on the bacterial community structure

The application of NP affected the relative abundance of different bacterial groups, but the effect depended mostly on the treatment applied to soil (Fig. 7). Only two bacterial phyla and two genera responded in the same way to the application of NP independent of the application of oats, earthworms or oats plus earthworms, that is, the relative abundance of Firmicutes and *Acetobacter* increased when NP were applied to soil and that of Verrucomicrobia and *Pedobacter* decreased.

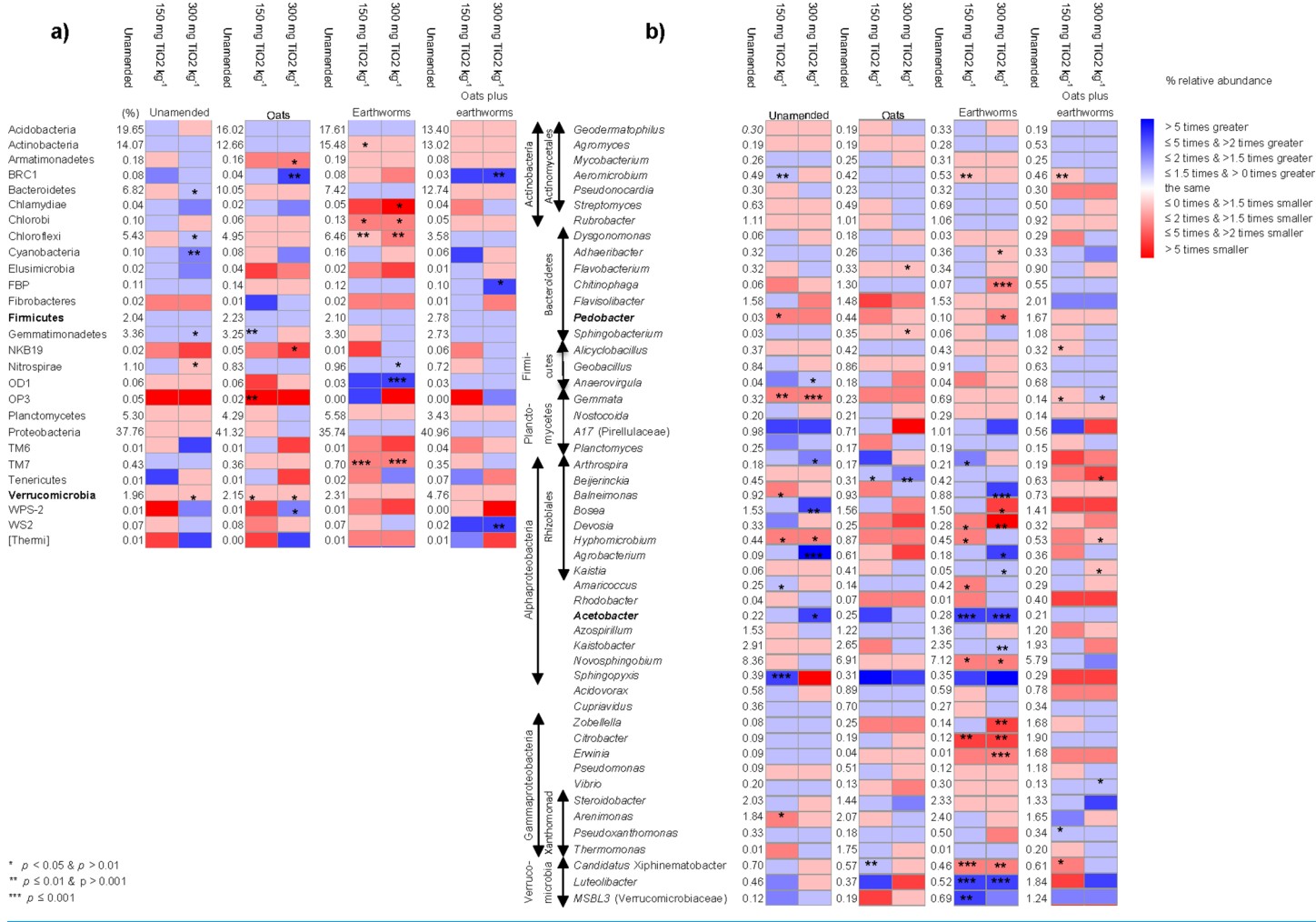

**Figure 7 The relative abundance of the different bacterial groups in unamended soil or soil amended with oats, earthworms or oats plus earthworms and heatmaps with the effect of the application nanoparticles when applied to the different treatments.** (A) Phyla and (B) Genera. Non parametric analysis with * $p < 0.05$ and >0.01, ** $p \leq 0.01$ and >0.001, and *** $p \leq 0.001$. The bacterial genera were grouped according to the bacterial phyla, class or order they belonged to. A red square means that the relative abundance of the bacterial group decreased when 150 or 300 mg $TiO_2$ nanoparticles were added to the unamended or soil amended with oats, earthworms or earthworms + oats compared to the soil not supplemented with $TiO_2$ and a blue square that it increased. The relative abundance of the bacterial groups (%) is given in the soil not amended with $TiO_2$ (Unamended), while the changes in the relative abundance in soil amended with 150 or 300 mg $TiO_2$ kg$^{-1}$ is given as the ratio of (the relative abundance in soil amended with 150 or 300 mg $TiO_2$ kg$^{-1}$)—(the relative abundance in unamended soil)/(the relative abundance in soil amended with 150 or 300 mg $TiO_2$ kg$^{-1}$).                               

The PCA analysis considering the different bacterial phyla showed an effect of incubation time on the bacterial community (Figs. 8A–8H). The PCA separated the soil with different application rates of NP toward the end of the aerobic incubation. In the earthworm-amended soil, for instance, the relative abundance of bacterial phyla, such as Chloroflexi, Firmicutes and TM7, was higher and that of Acidobacteria and Proteobacteria lower toward the end of the incubation than at the onset (Figs. 8E and 8G). The relative abundance of bacterial phyla, such as Actinobacteria, was higher when no NP were applied and that of Planctomycetes when 150 mg $TiO_2$-NP was applied toward the end of the incubation.

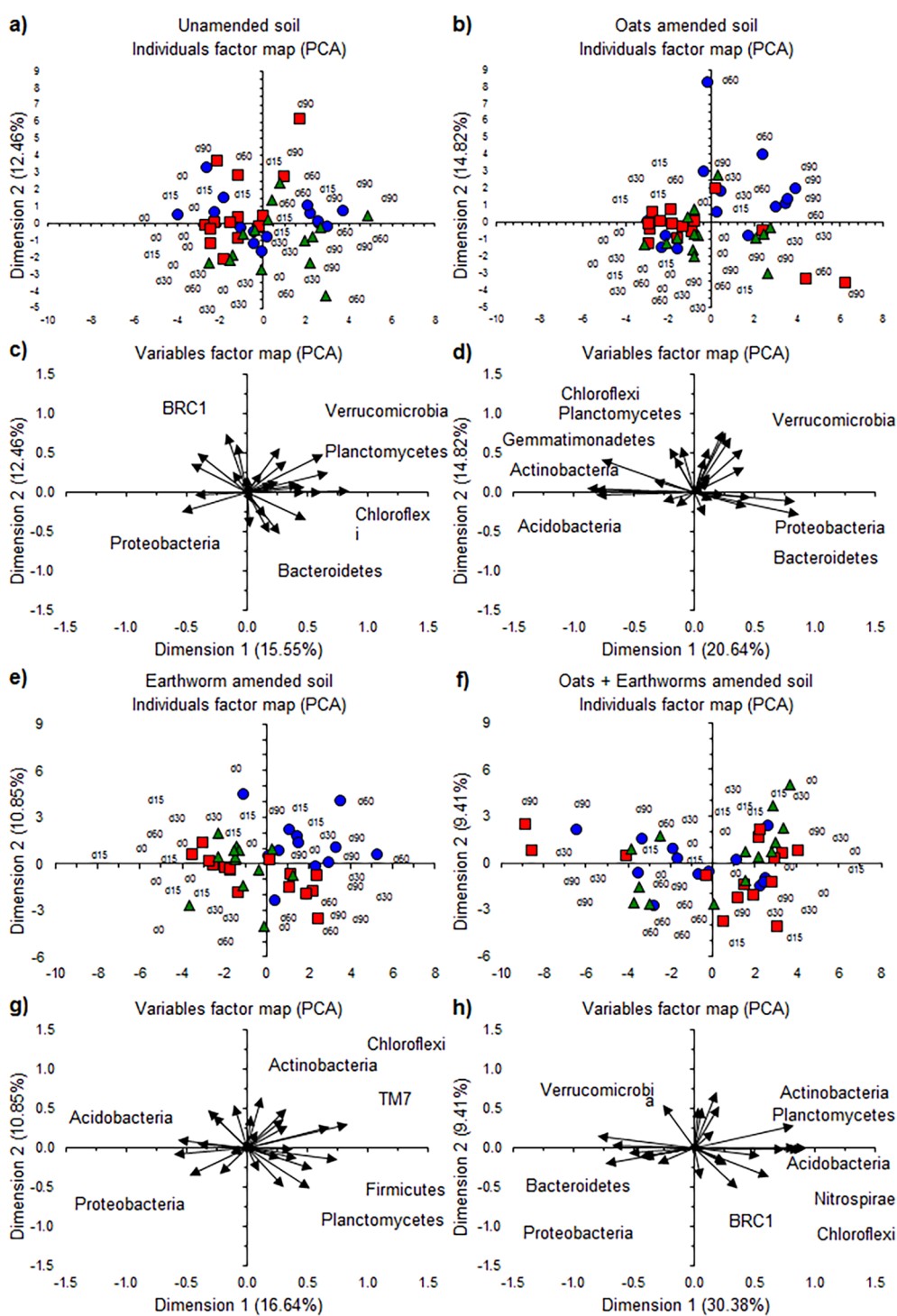

Amended with oats plus earthworms and left uncontaminated (●), contaminated with 150 mg nanoparticles kg⁻¹ soil (■), 300 mg nanoparticles kg⁻¹ soil (▲)

**Figure 8 The relative abundance of the bacterial phyla.** Individual factor maps and variables factor maps of the principal component analysis (PCA) of (A) and (C), the unamended soil; (B) and (D), soil amended with oats; (E) and (G), soil amended with earthworms (*Eisenia fetida* (Savigny, 1826)) and (F) and (H), amended with oats plus earthworms, respectively.

A PCA with the 50 most abundant bacterial genera showed an effect of NP when applied to the unamended and earthworm-amended soil, but not in the oats or oats plus earthworms amended soil (Fig. 9). Application of NP reduced the changes in the bacterial community structure over time as the relative abundance of members of bacterial genera, such as *Azospirillum* and *Cupriavidus*, increased more in the unamended soil than when TiO$_2$-NP was applied (Figs. 9A and 9C). Interestingly, the relative abundance of *Geobacillus* and *Thermomonas* increased more when 150 mg TiO$_2$-NP was applied to soil than in the unamended soil or when 300 mg TiO$_2$-NP was applied. In the oats-amended soil, the relative abundance of bacterial groups such as *Acidovorax*, *Chitinophaga*, *Flavobacterium* and *Pseudomonas*, increased over time while that of *Azospirillum*, *Flavisolibacter*, *Rubrobacter* and *Steroidobacter* decreased, but no clear effect of the application of NP emerged (Figs. 9B and 9D). Application of NP to the earthworm-amended soil altered the bacterial community structure over time (Figs. 9E and 9G). The relative abundance of genera, such as *A17* (Pirellulaceae), *Adhaeribacter*, *Citrobacter* and *Pedobacter*, increased over time when 150 mg NP kg$^{-1}$ were applied compared to the unamended soil, while in the uncontaminated soil the relative abundance of *Candidatus* Xiphinetobacter, *Gemmata* and *MSLB3* (Verrucomicrobiaceae) increased. In the soil contaminated with 300 mg NP kg$^{-1}$ changes in the bacterial community structure resembled those found in the uncontaminated soil, but they were smaller. Application of NP to the earthworms plus oats amended soil had no clear effect on the bacterial community structure (Figs. 9F and 9H). The relative abundance of genera, such as *Agromyces*, and *MSLB3* (Verrucomicrobiaceae) increased over time, while that of *Kaistobacter*, *Pseudoxanthomonas*, *Steroidobacter* and *Streptomyces* decreased.

The effect of NP on soil microorganisms is determined by the characteristics and amount of NP applied, soil characteristics, time of exposure to the NP and the treatments applied to soil. *You et al. (2018)* reported that zinc oxide NP decreased the number of bacteria in soil and changed the bacterial community structure in a saline alkaline soil, but other metal oxide NP, such as TiO$_2$-NP, cerium dioxide NP or magnetite NP, had a much smaller effect on the bacterial community. They related this to the fact that TiO$_2$-NP, cerium dioxide NP or magnetite NP were less soluble than zinc oxide NP under certain soil conditions. The amount of NP applied also affects its bioavailability and toxicity toward soil microorganisms (*Simonin et al., 2016*). For instance, *Ge, Schimel & Holden (2012)* reported that TiO$_2$ applied at 0.5 and 1.0 g kg$^{-1}$ had no effect on the bacterial communities after 15 and 60 days, but 2.0 g kg$^{-1}$ did. Not only the type of NP applied affects the soil microorganisms, but also the coating, particle size and phase composition determines the effect of NP (*Cornelis et al., 2014*). Consequently, the NP characteristics should be determined when their possible effect on microorganisms is investigated (*McKee & Filser, 2016*). Soil characteristics will also affect how NP alters soil processes and the microbial community (*Shoults-Wilson et al., 2011*; *Simonin et al., 2015*). Some types of metal oxide NP have been reported to affect the microbial biomass in some soils (*Dinesh et al., 2012*), but not in others (*McKee & Filser, 2016*). For instance, *You et al. (2018)* reported that zinc oxide NP decreased the number of Bacteria and changed the bacterial community structure in a saline alkaline soil, but not in a black soil. *Frenk et al. (2013)* reported

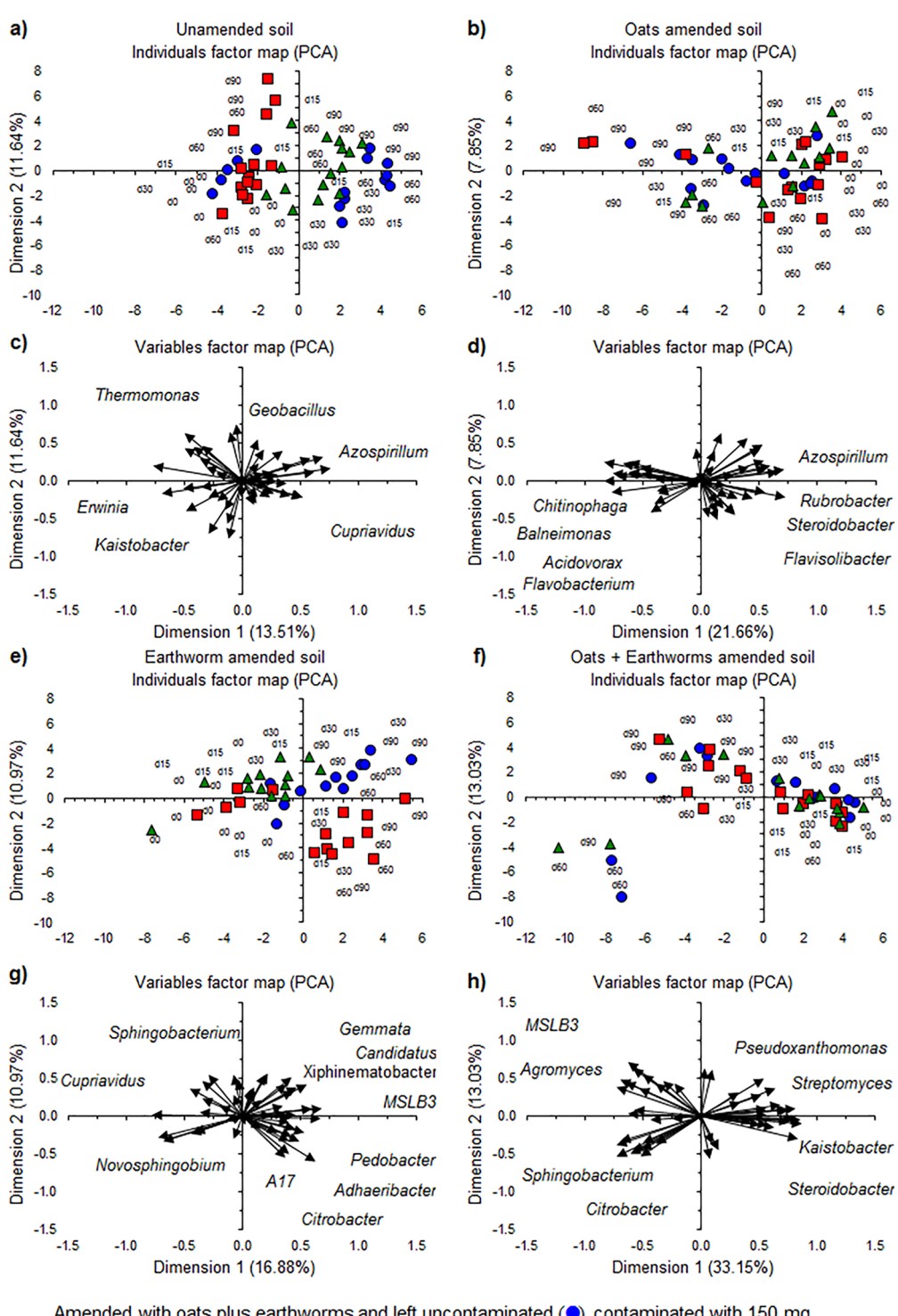

**Figure 9 The relative abundance of the 50 most abundant bacterial genera.** Individual factor maps and variables factor maps of the principal component analysis (PCA) of (A) and (C) the unamended soil; (B) and (D), soil amended with oats; (E) and (G), soil amended with earthworms (*Eisenia fetida* (Savigny, 1826)) and (F) and (H), amended with oats plus earthworms, respectively.

that the bacterial community in a sandy loam soil (Bet-Dagan) was more susceptible to the application of CuO and $Fe_3O_4$ nanosized particles (<50 nm) than that in a sandy clay loam soil (Yatir). Soil characteristics, such as aggregates formation and humic acids will affect bioavailability of the NP and thus its possible toxicity toward the microbial soil population (*Dinesh et al., 2012*). *Simonin et al. (2015)* reported that time of exposure to a NP also defined its possible effect. They found that $TiO_2$-NP applied at one or 500 mg $kg^{-1}$ soil had no effect on the bacterial community structure after 15 days, but it did have an effect after 90 days.

This study, however, clearly indicates that other factors also determine the effect of NP on the bacterial community structure, that is, earthworms, available organic material or a combination of both. In this study, the application of $TiO_2$-NP reduced the change in the bacterial community structure compared to the unamended soil over time and the effect was determined by the amount of $TiO_2$-NP applied. The application of $TiO_2$-NP reduced the change in the relative abundance of certain bacterial group and increased that of others. Application of $TiO_2$ also altered how the bacterial population changed over time in the earthworm-amended soil and the effect was controlled by the amount of $TiO_2$-NP applied. Application of organic material even in combination with earthworms negated the effect of $TiO_2$-NP applied on the bacterial population. So, the effect of $TiO_2$-NP on the bacterial population seemed to be linked to the amount of easily decomposable organic material. *Simonin et al. (2015)* reported that a possible effect of $TiO_2$-NP did not depend on soil texture but was controlled by pH and soil organic matter content. In this study, the easily decomposable organic material decreased in the unamended soil and even more so in the earthworm-amended soil as it was mineralized by microorganisms. Consequently, the effect of $TiO_2$-NP increased. Organic material forms complexes with NP, which reduces their bioavailability and thus their toxicity. Organic material mineralization will liberate the "fixed" NP. Application of organic material will negate that effect as it forms complexes with NP. It would be interesting to test that hypothesis and incubate a soil for an extended period of time and determine how the bioavailability of NP changes over an extended period, for example, a year, when organic material is applied. It can be hypothesized that organic material application would reduce the effect of NP, but with time the organic material would be mineralized and the effect of NP increase again.

Although the bacterial community structure was affected by the application of $TiO_2$-NP to the unamended and earthworm-amended soil, no such effect was found based on the functionality of the Bacteria. *Simonin et al. (2016)* reported that $TiO_2$-NP had a strong effect on soil microbial functioning by affecting the archaeal nitrifiers, but not on the bacterial nitrifiers. As microbial groups that are considered to be more resistant to stress, such as Archaea (*Valentine, 2007*), are affected by NP, other functions or other microorganisms might also be affected.

## SUMMARY

It was found that the application of oats and earthworms changed soil characteristics, but the application of $TiO_2$ NP had no effect on them and did not alter N mineralization. Application of oats as food increased the number of earthworms in soil as they reproduced,

while the application of $TiO_2$ NP did not affect them. The bacterial community structure in the unamended soil changed toward the end of the incubation as the amount of organic material that served as a C substrate for microorganisms decreased. The application of oats and earthworms altered the bacterial community structure. Application of oats increased the amount of C substrate available for the soil microorganisms while earthworms accelerated the decomposition of organic material by actively seeking it as food.

The application of $TiO_2$ NP had a limited effect on the bacterial community structure as only members of Firmicutes and Acetobacter were enriched while the relative abundance of Verrucomicrobia and Pedobacter decreased. The application of $TiO_2$ particles altered the bacterial community structure in the unamended and the earthworm-amended soil, but application of oats negated that effect. As such, application of organic material reduced the effect of the $TiO_2$ NP applied to soil, but as the organic material was mineralized by the soil microorganisms, the effect of $TiO_2$ NP increased again after some time.

## ACKNOWLEDGEMENTS

We are grateful to de León-Lorenzana A. S. and Ramírez-Barajas B. E. for laboratory support and Prince L. for the language review.

### Funding

This research was funded by "Consejo Nacional de Ciencia y Tecnología" (CONACyT, Mexico) 'Ciencia Básica SEP-CONACyT' projects 151881 and 287225 and Cinvestav (Mexico) while Katia Berenice Sánchez-López received grant-aided support from CONACyT. The funders had no role in study design, data collection and analysis, decision to publish, or preparation of the manuscript.

### Grant Disclosures

The following grant information was disclosed by the authors:
"Consejo Nacional de Ciencia y Tecnología" (CONACyT, Mexico) 'Ciencia Básica SEP-CONACyT' projects 151881 and 287225.
Cinvestav (Mexico) while Katia Berenice Sánchez-López received grant-aided support from CONACyT.

### Competing Interests

The authors declare that they have no competing interests.

### Author Contributions

- Katia Berenice Sánchez-López performed the experiments, analyzed the data, authored or reviewed drafts of the paper, approved the final draft.
- Francisco J. De los Santos-Ramos performed the experiments, approved the final draft, chemical analyses.
- Elizabeth Selene Gómez-Acata performed the experiments, analyzed the data, authored or reviewed drafts of the paper, approved the final draft.
- Marco Luna-Guido performed the experiments, authored or reviewed drafts of the paper, approved the final draft, chemical analyses and obtaining and manipulating the soil samples.
- Yendi E. Navarro-Noya analyzed the data, contributed reagents/materials/analysis tools, prepared figures and/or tables, authored or reviewed drafts of the paper, approved the final draft.
- Fabián Fernández-Luqueño conceived and designed the experiments, analyzed the data, contributed reagents/materials/analysis tools, authored or reviewed drafts of the paper, approved the final draft.
- Luc Dendooven conceived and designed the experiments, analyzed the data, contributed reagents/materials/analysis tools, prepared figures and/or tables, authored or reviewed drafts of the paper, approved the final draft.

## Data Availability

All sequences are available in NCBI Sequence Read Archive (SRA) under the BioProject accession number PRJNA453453.

## Supplemental Information

Supplemental information for this article can be found online at http://dx.doi.org/10.7717/peerj.6939#supplemental-information.

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
