# Peer review of "TiO2 nanoparticles affect the bacterial community structure and Eisenia fetida (Savigny, 1826) in an arable soil"

_PeerJ, doi:10.7717/peerj.6939_

## Round 0.1 · original submission · Major Revisions

Please see the comments made by the Reviewers and consider revising your manuscript according to their suggestions. In addition, please consider rephrasing the title of the manuscript so that it would not pose a question that could be answered “yes” or “no”, because, I believe, published literature has provided evidence that the answer to this questions is “yes”. In your abstract you pose the research question as: “how nanoparticles might interact with earthworms and organic material and how this might affect the bacterial community structure and their functionality”. I suggest using “how” question in the title if you prefer the title to be in a question format.

Additionally, please modify Figures 1 and 2, so that statistically significant differences (if any) would be indicated for the data presented.
I look forward to reading your revised manuscript!

Reviewer 1 ·

Basic reporting

No comment

Experimental design

No comment

Validity of the findings

No comment

Additional comments

This study focused on the effects of nanoparticles on bacterial community structure. Also how the presence of earthworms, oat seeds or their combinations affect soil properties and bacterial community were examined. This topic is interesting, which is helpful to understand nano-toxicity in the environment. Overall, this study is comprehensive, the experimental design and data can support the conclusion. The writing is also easy to follow. Some major issues need to be addressed before accept.
1) Please rewrite the abstract. Now it is hard to find the novelty of this work through the abstract. Some main and important results are very vague. For example, the last sentence, I think if we did not do this study, we still have this kind of conclusion. So what are your special results in your study?
2) Results and Discussion. The title of this part such as soil characteristics, earthworms, unamended soil et al. needs to be revised. The whole sentence is better to summarize the following results.
3) Conclusion part needs to be rewritten. This part looks like an abstract since much specific results are shown in this part.
4) The writing needs to be carefully checked. There are many gramma mistakes found in the manuscript. Probably a professional language editing company is helpful.

·

Basic reporting

The language is clear and professional. The raw data of the microbial community analyses are provided in an online repository.
A weakness of the manuscript is the quality and labelling of the Figures and Tables which could use improvement. All graphs are quite small and the quality is low which makes it hard to read labels and axis names. The symbols chosen in Figure 1 are 2 could be grouped together for the different treatments e.g. all earthworm treatments are black and different shapes; all earthworm+oats treatments are white and different shapes.
Figure 1: The letter a)-l) are easily overseen, they should be larger and placed at the right of each graph. Differences between treatments in a) and b) are hard to see because of the scale chosen. Maybe a Table would show these more clearly.
Figure 3: This is very crowded and therefore hard to read.
Figure 4 and 8: It is not clear what the three columns depict. In Figure 8 it is not clear what the arrows show. Please add information to clarify this.
Supplementary information
Descriptions and/or legends of the Figures and Tables are needed otherwise it is difficult to understand what is shown.
Table 4: This Table should be moved to the main manuscript.

Experimental design

The authors have studied an important and interesting subject by investigating the effects of the combination of TiO2NP, earthworms and organic matter on a soil microbial community. The novel aspect of this study is the investigation of effects of combinations of different treatments (TiO2NP, earthworm presence, organic matter addition) on a soil microbial community. The Introduction could emphasize this more by pointing out that most studies focus on effects of changing a single parameter (e.g. addition of nanoparticles or adding earthworms).The objective of the study is clearly stated. Appropriate methods were applied for investigation of the research objective. The description of the experimental design could, however, be improved in parts by giving more details.
l. 83 It would make it easier for the reader if the basic characteristics (e.g. size) of the used NP were given here, rather than only in the Supplementary Information. Also information on the coating or stabilizing agent of TiO2NP is missing. Numerous studies have shown that this can strongly affect behavior and toxicity of NP (e.g. Tan et al. 2017, Environmental Pollution; Li et al. 2013, Environmental Science & Technology; Whitley et al. 2013, Environmental Pollution; Arnaout et al. 2012, Environmental Science & Technology).
l. 91: What kind of kitchen organic waste was fed to the earthworms? Is there the possibility that the earthworms were pre-exposed to e.g. pesticide residues in the organic waste? If this was the case their sensitivity to other chemical stressors might differ from animals that were not pre-exposed.
l. 95: Where pesticides and fertilizers applied to this field?
ll. 101: To increase readability this data could be presented in a Table. Was this data gathered as described in 2.8? If so, please refer to this section.
l. 105: The heading should be more detailed to guide the reader better: “Vessel preparation and experimental design”.
l. 105-109: It is not clear why this procedure was chosen and with which goal.
l. 110-116: It would be easier to understand which treatments were applied by inserting a Table here rather than in the Supplementary Information. How many individuals of E. fetida were used per treatment? What amount of oat seeds was added per treatment? How many replicates were done per treatment? Are these concentrations of TiO2NP environmentally relevant or why were they chosen?
l. 120: It is not clear what the difference between “treatment” and “soil sample” is. How many replicates were measured per time point?
l. 171-175: Would it be possible to create a models with NP, earthworms, oats to explain soil characteristics and earthworm “characteristics”? This way the statistical power would be increased compared to doing multiple t-tests.
l. 174: 2-way t-test
ll. 175: Because of the importance of PCA and CAP analyses for this paper it would be helpful for readers who are less experienced in reading such graphs to briefly explain how to do so.

Validity of the findings

The Results and Discussion section is very long and could benefit from shortening in parts. This way the key findings could be more clearly identified.
The Conclusion is a summary of the findings of the study rather than a Conclusion on how this study has advanced our understanding of interactions in soils.

---

## Round 0.2 · Minor Revisions

Thank you for carefully revising your manuscript. I have reviewed your rebuttal letter and the revised manuscript and would like to ask you to make a couple of additional changes before I can either make a decision or send the manuscript to reviewers.

In your rebuttal letter you mention adding Table S3 with the statistical analysis of data depicted in Figures 1 and 2. The table did not appear in the editorial system. Could you please double-check and upload the table?

Also, I noticed, in your response to Reviewer 2 comment: "ll. 175: Because of the importance of PCA and CAP analyses for this paper it would be helpful for readers who are less experienced in reading such graphs to briefly explain how to do so." there were no edits made in the manuscript. Please see if this comment can be addressed in the manuscript and if figure captions can be made more informative.

Please also check that each panel (labeled as a, b, c etc) in the figures is described in the caption. For example, in Figure 1 there are two graphs for each parameter shown (for example, a and b are both pH changes, but it is unclear what is the difference between a and b, and similarly for each of the pair of graphs for other parameters).

I appreciate your hard work and look forward to hearing from you soon.

---

## Round 0.3 · Minor Revisions

Dear authors,
Please make appropriate changes to the manuscript according to Reviewer 2 suggestions.

Reviewer 1 ·

Basic reporting

no comment

Experimental design

no comment

Validity of the findings

no comment

Additional comments

no comment

·

Basic reporting

No comment

Experimental design

No comment

Validity of the findings

No comment

Additional comments

The authors have clearly improved the quality of the manuscript with their revisions.
There are still a few points that should receive more attention.
1) The authors’ response stated that pesticides were only applied to the soil to a very limited extent. This information should appear in the manuscript as well.
2) Possible accumulation of TiO2NP by earthworms is now mentioned in the manuscript, however a reference is missing. References are also missing in ll 474.
3) The Conclusion was changed, however it still mainly is a summary of the data. If this is the goal the heading should be changed to Summary. Otherwise this section needs additional work.

---

## Round 0.4 · accepted · Accept

There are no additional comments.

#